# Quality of Life and Burden of Disease in Italian Patients with Transfusion-Dependent Beta-Thalassemia

**DOI:** 10.3390/jcm11010015

**Published:** 2021-12-21

**Authors:** Fabio Tedone, Piero Lamendola, Stefania Lopatriello, Davide Cafiero, Daniele Piovani, Gian Luca Forni

**Affiliations:** 1Helaglobe SRL, Via L. da Vinci, 16, 50132 Firenze, Italy; stefania.lopatriello@helaglobe.com (S.L.); davide.cafiero@helaglobe.com (D.C.); 2Istituto Clinico Humanitas IRCCS, Via A. Manzoni, 56, 20089 Rozzano, Italy; d.piovani@hotmail.com; 3Ematologia Centro della Microcitemia e delle Anemie Congenite-E.O. Ospedali Galliera, Mura delle Cappuccine, 14, 16128 Genova, Italy; gianluca.forni@galliera.it

**Keywords:** β-thalassemia transfusion-dependent, health-related quality of life, psychological well-being, iron chelation therapy, burden of disease, physical health, psychological health, SF-36, PGWBI

## Abstract

Lifespan treatment in transfusion-dependent β-thalassemia (TDT) is expected to impact quality of life. This study aimed at evaluating health-related quality of life (HRQoL), well-being, and the burden of TDT on Italian patients. Patients (≥14 years) were invited to complete a cross-sectional, online volunteer survey. HRQoL was measured by the 36-item short-form health survey (SF-36) and wellbeing was measured by the Italian version of the Psychological General-Well-Being-Index (PGWBI). A total of 105/167 completed questionnaires were analyzed (46% males; median age 44, (IQR = 11)). Patients reported lower HRQoL compared with the general population in all SF-36 domains (except for emotional well-being (*p* = 0.7024) and role limitations due to emotional problems (*p* = 0.1389)). PGWBI domains general health and vitality and the total PGWBI score were all significantly lower (*p* = 0.0001) compared with the general population. On average, patients spent 16.62 h/month engaged in care activities that were additional to the time required for completing transfusions. Of the 16.62 h/month, 11.7 h/month were required for therapy management and 4.92 h/month for family management. This study found lower HRQoL and well-being in physical and psychological domains compared with the general population. Further, patients were found to have a high perceived burden of disease.

## 1. Introduction

Transfusion-dependent β-thalassemia (TDT) is a chronic, hereditary disease characterized by severe anemia. More than 300 β thalassemia mutations have been studied, differentiated by a severity index; β+ refers to mild mutations causing a relative reduction of β-globin chain synthesis and β0 refers to severe mutations leading to a complete absence of β-globin chain product [1]. Deficient production of β-globin chains results in ineffective erythropoiesis and premature hemolysis of circulating red cells. Ineffective erythropoiesis leads to decreased production of the hepatic hormone hepcidin, which in turn results in increased intestinal iron absorption [1]. The disease is diagnosed within the first few years of life, and standard treatment involves repeated blood transfusions throughout life.

Treatment is also often combined with supportive and stabilizing medications, called chelating agents, to reduce the presence of the iron in the blood. TDT is the most severe form of the disease. Although modern medical treatments have reduced mortality and morbidity associated with patients with thalassemia [2,3], the life expectancy is still lower (80% of patients have a life expectancy of more than 40 years [3]) than the general population (82 years [4]). In addition, the need for continuous treatment throughout the course of one’s life, the daily intake of chelators, and the need to constantly monitor the levels of iron in the blood are all expected to negatively impact the health-related quality of life (HRQoL) of patient with β thalassemia. HRQoL is a multidimensional construct related to the specific impact of the disease on quality of life. It consists of three broad domains (physical, psychological, and social functioning) that are conditioned by the disease and/or treatment. Physical functioning refers to the physical symptoms caused by both the disease or those related to treatment, and to the individual’s ability to carry out daily activity. The psychological functioning (which can also include cognitive functioning) varies from great discomfort to a positive sense of well-being. Finally, social functioning refers to the qualitative and quantitative aspects of the social integration of relationships and interactions [5]. In general, significant progress has been made in understanding the impact that chronic disease can have on patients and their lives, and several studies have been carried out in order to assess the quality of life in patients with thalassemia [6]. However, in the Italian context, the literature is sparse. To the best of our knowledge, the latest data were published in 2008, from a study carried out by Scalone et al. [7]. In this Italian study, 137 patients from eight different centers were studied, and both the physical and psychological components of HRQoL were found to be impacted. Also in Italy, using the SF-36, a study [8] interviewed the adult population first in 2001, and then again in 2009, to identify any changes in the quality of life. Although the patients showed an improvement in mental domain of HRQoL in the second measurement, both in 2001 and in 2009, the scores of people with thalassemia were lower in all other domains compared with the general Italian population. Of note, the mentioned study surveyed patients resident in Liguria, an Italian region. Therefore, these results may not be representative of the general Italian thalassemic population. Well-being is also an important part of HRQoL assessments. More literature has explored measuring well-being, because it is a validated population outcome for the assessment of how people perceive their life [9,10,11,12,13]. It is associated with a lower risk of disease, better immune system functioning, faster recovery, and greater longevity [14,15,16,17,18]. It is also a good indicator of people’s functions [19]; that is, individuals with high levels of well-being are in fact more productive at work and are more likely to contribute to their communities [20]. Finally, it is an independent predictor of mortality in different patient populations (as well as in the healthy population) [21,22]. Therefore, the aims of our study are as follows:Evaluating the disease impact on HRQoL and well-being of TDT patients;Assessing the burden of disease in TDT patients.

## 2. Materials and Methods

TDT patients aged ≥14 years were invited to a cross-sectional, volunteer-based, online survey (link created on the SurveyMonkey site), circulated by four patients’ associations (United Onlus, Fondazione Giambrone, Alt Rino Vullo Ferrara, Associazione Veneta Lotta Alla Talassemia). Patient participation was preceded by patient informed consent; because of the nature of the research, the study did not require gaining ethical approval and it followed UNITED’s compliance rules for communication with patients. The questionnaire was structured as described below:Section 1—Socio-demographic information: age, gender, region of birth, family group description, domicile, education level, working status;Section 2—QoL assessment: SF-36 (36-item short-form health survey) questionnaire;Section 3—QoL assessment: PGWBI (Psychological General Well-Being Index);Section 4—burden of disease assessment:
(a)Diagnosis and center care features(b)Transfusion programme(c)Iron chelation therapy(d)Monthly disease treatment activities: (1) time spent actively involved in transfusions and iron chelation thereapy and related perception on daily living; (2) time spent in actions preparing for the transfusion and iron chelation therapy and related perception on daily living.

Section 1 and Section 4 were drafted by the authors and are based on literature research; the Thalassemia International Federation 2014 guidelines informed the questionnaire as to the pattern of care [23]. HRQoL was measured by the Italian validated version of the generic instruments SF-36 questionnaire and PGWBI questionnaire [24]. Sample size was estimated based on literature; as reported in Gandek, 1993 [25], 90 SF-36 questionnaires are sufficient to detect at minimum 10 points differences between two groups. Concerning the PGWBI index, papers assessing PGWBI on patients affected by TDT are not available, but studies reported significant differences between groups with samples smaller than 60 individuals [26,27]. Two-sample T-test for comparisons was applied (*p* < 0.05). Comparisons of continuous variables with a reference value were analyzed via one-sample T-test. All statistical analyses were performed using Stata 16 (Stata Corporation, College Station, TX, USA).

## 3. Results

### 3.1. Sample Feautures

In total, 167 TDT patients, all receiving regular transfusions based on the prescription of their center, participated in the survey. Of these, 105 (62.87%) complete questionnaires were analyzed; the remaining 62 questionnaires were excluded because patients completed only Section 1. Further, 48 patients (46%) were male and the average sample age was 42.4 ± 9.16 years, with the youngest patient aged 14 years and the oldest 64 years. Around 70% of the patients were in the age range of 35–54 years (Figure 1).

TDT was diagnosed in childhood at the age of 1–2 years (79%); 98% of respondents were treated for over 10 years; bone marrow transplant are reported in only 1.9% of respondents. Approximately 50% of respondents live in a household with their partner of family. Furthermore, 35.2% of patients lived on the Islands and 27.6% lived in the North (Lombardia, Piemonte, Valle D’Aosta, Liguria, Trentino Alto Adige, Veneto, Friuli Venezia Giulia). Over 75% of respondents had either high school or university education. Around 28% of respondents are unemployed or unable to work (Figure 2). Around 85% of patients suffers from one or more comorbidities; the most frequently reported comorbidities were osteoporosis (74.28%) and endocrine problems (54.28%) (Figure 3).

### 3.2. Health Related Quality of Life and Well-Being 

Italian TDT patients reported significantly reduced levels of HRQoL compared with the general population in all the SF-36 domains, except in the categories of emotional well-being and role limitations due to emotional problems (Table 1). These findings reflect that patients suffer limitations in social activities (like meeting friends), physical functioning (like walking, dressing), bodily pain, role (like working activities), and energy/fatigue.

When assessing the level of psychological well-being by means of PGWBI index, Italian TDT patients reported significantly lower levels when compared with the general population (Table 2). The total mean PGWBI score was 71.91, which is significantly lower (*p* < 0.0001, IC 68.46–75.36) than the mean PGWBI total score for the general population (77.8). Additionally, this score indicates moderate distress [24]. Significant (*p* = 0.0001) differences were also reported in the general health and vitality domains.

### 3.3. Pattern of Disease Management

The questionnaire could not specifically ask patients if they were treated at transfusion centers working in a network, i.e., SITE (Società Italiana Talassemie e Emoglobinopatie, the Italian Society of Thalassemia and Hemoglobinopathies), because the respondents could not be aware of that. Therefore, the questionnaire asked for any knowledge about the center of care the respondent might have. Patients reported that they are treated in specialized daily centers of care for hemoglobinopathies (around 60%), while 21% of patients are treated in transfusion centers and 11.4% in hematology wards such as the outpatient setting located within hospital wards (these are transfusion rooms within hematology or internal medicine wards). The remaining 5% of patients are treated at any ‘other’ type of center (specialized day service, internal medicine ward), while 2.9% of patients are not aware of the type of center at which they receive treatment. For 87% of patients, the center of care is located at their place of residence. The geographic distribution of the centers from our sample is reported in Figure 4.

### 3.4. Iron Chelation Therapy

Oral chelation is the most prescribed iron chelation therapy (80%), followed by subcutaneous therapy (13.33%) either in isolation or combined with infusion therapy (3.8%). Iron chelation treatment is mostly monitored at transfusion centers. Complications of iron overload affect 30% of patients and they are mostly endocrinological disorders (Figure 5). Of the participants with complications, 27% of them are affected by two or more complications. The mean score of perceived burden due to the management of iron overload complications is 5.76 ± 3.11 (1 = minimum, 10 = maximum). In particular, patients following only the oral therapy reported a lower burden (5.32 ± 3.06) compared with those patients following an isolated or combined subcutaneous therapy (7.74 ± 2.64). The results are statistically significant as confirmed by the two-sample T-test (*p* = 0.0015). As a matter of fact, the type of chelating therapy did not affect the quality of life and psychological well-being (no statistically significant differences were found for neither the PGWBI nor the SF-36 domains).

### 3.5. Burden of Disease

Patients reported a high burden from their transfusion program (average score 6.65 ± 2.68, 1 = lowest burden, 10 = highest burden). As reported in Figure 6, the reasons for perceived burden are more related to the chelation therapy, causing fear or discomfort in approximately one-quarter of respondents, because of complications from iron overload and burden of therapy. A lower proportion of patients consider transfusion schedules (for instance, frequency of transfusions or distance from care centers) and the short-term transfusion-related adverse events as a burden.

A total of 70% of patients follow a transfusion program; however, only 45% adhere to the scheduled program. The reasons for not being compliant with the transfusion schedule in the remaining 55% of cases were mostly related to care center problems such as blood bag availability, changes in timetables (69%), and clinical reasons (hemoglobin value variations, 31%). Most of the patients feel supported by their own center of care especially when they receive assistance by a multidisciplinary team (Figure 7). Patients were also asked if their center supports them in managing their comorbidities (Figure 8).

It was recorded that support is limited to disease, which is usually charged by the National Health System (dental and fertility disorders are generally out-of-pocket expenditure for most of Italian citizens). However, around 80% of respondents reported centers organizing routine check-ins, periodical check-ins, and blood tests. These mostly occurred on the same day as transfusion (68%). Transfusion centers also support patients in the case of transfusion adverse events, which cause fear in patients (average score 6.82 ± 2.91, 1 = minimum, 10 = maximum), reported from 59% of respondents.

Over 94% of TDT patients missed at least 1 day of work/school per month and 43% had to reduce their working time or change their job due to TDT (Figure 9). Moreover, 78% of RBC transfusions are provided during the morning.

### 3.6. Monthly Disease Treatment Activities: Non-Transfusion and Transfusion Time

Time spent for the TDT-related activities during both transfusion and non-transfusion days over one month was investigated. Patients spent, on average, approximately 16.62 h per month for transfusion-related activities during non-transfusion days (Table 3).

As depicted in Figure 10, the demanding time requirements of transfusion-related activities on non-transfusion days can have an impact on daily life activities and time taken from family and work.

Patients reported that time spent on transfusion days was related to reaching transfusion centers, waiting to start the transfusion session, implementing the transfusion session, or waiting for specialist visits (Figure 11).

## 4. Discussion

We hypothesized that TDT, transfusion dependency, and its related iron chelation therapy would affect patients HRQoL in both physical and psychological domains, and cause a high perceived burden of disease due to lifespan treatments. We also hypothesized that Italian TDT patients may have better outcomes of treatment at their transfusion centers, because most Italian transfusion centers work in the network of SITE, with common protocols and standard tools. However, we wanted to assess this in the real setting of care. This objective led us to implement a study design that could directly give voice to TDT patients, without any intermediated selection from clinical centers. This was achieved through voluntary participation from patients, using an online survey tool disseminated through their patients’ associations. This represents an absolute novelty in the literature on this subject. Another influencing factor for the study method, deriving from the assumption of well treatment of TDT patients in Italy, was that their HRQoL should have been compared with the general population to assess possible diversity. In doing so, this study was also novel in its usage of general instruments of assessment of HRQoL, instead of specific tools like TranQoL, i.e., the SF-36 scale and the PGWBI index, which are generic and not a verified disease-specific QoL instrument for TDT. These scales are complementary tools that assess diverse elements of mental domains. Finally, for the first time in the literature, our study assessed the burden of disease over a month of TDT patient daily life by mirroring their pattern of care derived from international guidelines and scoring their burden of care using the Likert scale from 1 (minimum) to 10 (maximum). Based on 105 complete questionnaires, which represented the target threshold of sample validity for the HRQoL assessment, we measured lower levels of HRQoL compared with the general population in all of the SF-36 domains, excluding emotional well-being (*p* = 0.7024) and role limitations due to emotional challenges (*p* = 0.1389). In addition, well-being as assessed using the PGWBI score showed lower overall psychological well-being (*p* = 0.0001) and a significantly lower level of well-being in the general health and vitality domains (*p* = 0.0001). Through these assessments, well-being was defined as a moderate level of distress. General health and energy/vitality domains were impaired in both the SF36 and PGWBI index. This likely means that patients experience less energy to cope with daily ordinary and extraordinary life, as well as slowness in executive function and quality of sleep. The burden of disease is high in terms of time spent for transfusions and chelation treatment; patients spent a mean of 16.62 h/month for non-transfusion activities, due to therapy management (managing specialists visits, blood tests, therapy complications) and family management (time subtracted to family activities, babysitting, organizing work permissions). Additional activities during the transfusion day account for an additional 6.64 h/month, which included transport to transfusion centers and waiting for and receiving RBC transfusion.

This paper represents a relevant and necessary update on HRQoL assessment in TDT Italian patients. Scalone et al. demonstrated in 2008 [7] that both physical and mental domains of HRQoL measured by the SF36 scale were compromised compared with the general population. We confirmed this result in all of the SF-36 domains (*p* < 0.05), except for emotional well-being (*p* = 0.7024) and role limitations due to emotional problems (*p* = 0.1389). However, when comparing SF-36 data from our sample with the Scalone et al. sample, our study shows a further impairment (with respect to 2008) in the domains of physical functioning (*p* = 0.0012), energy/fatigue (*p* < 0.0001), social functioning (*p* = 0.0453), and general health (*p* < 0.0001) (please see Appendix A, Table A1). In Scalone et al., patients were longitudinally enrolled between November 2005 and March 2006 from thalassemia treatment centers they attended. In our study, patients associated with patient advocacy organizations voluntary participated in the survey and are not all treated at dedicated, specialized TDT centers involved in a network, like the aforementioned study. Hence, our sample is more representative of the TDT population where patients can be treated in any type of center, not those that are only specialized. HRQoL assessments allow the ability to identify more significant differences with the general population when assessed in a real setting of care. An additional difference between the two studies is related to the sample age, which is younger (28 years) in Scalone et al. This might have impacted the outcome of HRQoL related to the duration of TDT treatment follow-up. Our study HRQoL results can be considered robust as far as the sample size estimation was concerned, because we collected a sufficient number of complete questionnaires, based on the literature [25,26,27]. Another aspect of robustness is the use of the different but complementary tool of HRQoL assessment. Improvements have been made in the treatment of TDT, but HRQoL is increasingly of interest. It is important how a disease or a treatment is experienced by the patient [28] in terms of physical, psychological, and social functioning of HRQoL. To do this, we used SF-36, an internationally validated and well supported tool. It has been scientifically validated in Italy [29], and used in assessing HRQoL in the general population and population with diseases [30]. Furthermore, the tool has been used in several studies assessing HRQoL of patients with TDT [6]. More relevant, previous Italian HRQoL results regarding TDT are uncertain, because they show prominent impairments in social functioning, role-emotional, and mental component, as assessed by the symptom-check-list-90 revised (SCL-90-R) in Messina et al. [31], or show scores almost close to the country norms, as in the mental domains of SF-36 in Scalone et al. We thus hypothesized that there exists a compromised mental HRQoL in TDT patients, and we proposed that SF-36 was not appropriate to assess this domain. Thus, we used an additional tool to investigate the mental impact in TDT, while taking a conservative position that did not consider TDT patients’ mental health as compromised as in psychiatric patients. We thus selected the PGWBI well-being scale, already validated in Italy [24]. In fact, the literature shows that a positive mental state is much more than the mere absence of symptoms, and that it is necessary to include the measure of well-being in studies that measure health outcomes and quality of life [32]. The SF-36 is used to measure health status, plan services, and measure the impact of clinical and social interventions [33]. Conversely, the PGWBI is a specific tool for assessing subjective psychological well-being in the general population [34]. These appropriate choices of instruments allowed us to more deeply and precisely assess the existence of mental impairment in TDT patients. We measured impairments in patients’ well-being, even if limited to general health (*p* = 0.0001) and vitality (*p* = 0.0001) domains, confirming the results from the SF-36 scale (*p* = 0.0001) in both domains. However, these domains have slightly different clinical significance in these two scales. Vitality in SF-36 refers to aspects of feeling of energy, while vitality in PGWBI contemplates aspects such as slowness in executive function and quality of sleep [35]. Finally, PGWBI puts great emphasis on personal psychological well-being [34], while the general health subscale in SF-36 is strongly related to physical health measures [36].

Overall, the PGWBI scores are significantly lower than the general population (*p* = 0.0001), demonstrating that TDT patients are affected by moderate distress compared with the general population [24]. Overall, we can affirm that patients are treated as far as the immediate life-threatening anemia by means of transfusions and related iron chelation therapy, but their levels of distress, well-being, and physical dysfunction can be compromised, despite appropriate care and persons’ capability to manage and cope with usual and unusual daily life events.

This work is also the first to assess the burden of disease in TDT Italian patients’ daily life. The total time spent for transfusion (median value 8.57 min/day) is in the range of the value reported in Paramore et al. [37] (median value 10 min/day), where data from 85 patients, mostly from USA and the United Kingdom, were analyzed. There were differences related to insurance payment activities, a significant time component in Paramore et al. [37]; this factor is missing in the Italian NHS, which is free of charge for Italians. Regarding the time range spent for each activity during transfusion days, this study reports that, on average, an additional 6.64 h/month are spent for each transfusion. Again, the result (9.8 h/month) is similar to that reported in Paramore et al. [37] (9.8 h/month), where the main difference is again the time spent for insurance payments (2.53 h/month). Finally, the immense burden of disease is illustrated by the fact that one-quarter of patients are unemployed due to thalassemia and in fact all of them lost at least 1 day/week during work/school time for transfusion, mostly occurring during the morning.

Sociodemographic features align with the most recent Italian data from 1899 patients [38]. The gender distribution was balanced between males and females, as in Conte et al. [38] (nearly 47% males in both samples). Our sample is older, 42.43 ± 9.16 years versus 30.20 pm1.0 in this database [38], likely owing to the fact that it collected data also from the pediatric population. Age distribution is similar, with around 73% of patients in the range of 25–45 years in the database and 88% in the range of 25–54 years in our study. The oldest patient was 64 in our study and 65 years old in the database. The prevalence of TDT carriers in Italy markedly differs, with some regions historically having a higher prevalence than others. This is especially true in Sardinia, Sicily, Southern Italy, and the Po-delta area; the geographical distribution of birth place (and domicile) follows this pattern in our sample [39]. As far as education level and employment, the results are also comparable to previous Italian data from a 150-patient survey, from which 50% of patients have a high school degree, around 50% are employed (including self-employment), and around 20% are unemployed [40]. Complication rates are aligned with the literature; that is, TDT is associated with a marked increase in the risk of endocrine disorders, including osteoporosis, hypogonadism, thyroid and parathyroids, and glucose metabolism disorders. Even in presence of effective iron chelation and optimal transfusion scheduling, the prevalence of these disorders is high [41]. TDT patients should receive treatment in “specialized thalassemia centers”, i.e., clinical settings and sites where appropriate disease management tools, including specialized staff and transfusion center able to provide personalized assistance and treatment, are available, as defined by SITE protocols for networking referral centers [40]. The literature demonstrated that therapeutic centers and services in Italy are extremely heterogeneous in terms of the age of patients (adult versus pediatric), typology (clinical settings versus transfusion settings), size (from very small to very large), and range of services offered to the patients (medical expertise, laboratory services, and iron overload monitoring tools) [38]. As a matter of fact, the HTA-Thal database reports that Italian TDT patients are referred to a variety of clinics (adult, pediatric, or transfusion centers or units). This picture also resulted from this study where only nearly half of patients are not followed at dedicated day hospital centers. Our geographical distribution of centers follows the distribution of the HTA-Thal database. Half of patients felt supported by their center of care and better supported if a multidisciplinary team was available at the center. This is aligned with the requirements of Thalassemia International Federation Guidelines for the clinical management of thalassemia, stating that “the unit should be dedicated, should include physician experts in thalassemia and should be linked to other specialists and services as indicated” [38]. Indeed, being treated in specialized centers for hemoglobinopathies increased survival rates of patients with thalassemia major (*p* < 0.0001 versus patients treated in non-specialized centers) [41]. On the other hand, two main factors affect survival rates in TDT patients: the lack of adequate blood transfusions (although this is particularly relevant in developing countries) and complications due to iron overload [42]. In fact, in regular blood transfusions, iron overload can lead to significant morbidity of the heart, liver, and endocrine glands [42]. This is also in alignment with our results, as the same complications from iron overload affect TDT patients. As far as oral chelation therapy, it is more represented in our study compared with other publications (where the average oral prescription rate is around 50%). However, this result is aligned with the fact that oral agents, such as deferiprone and deferasirox, are generally preferred because of a convenient route of administration associated with better compliance, as well as lower serious adverse effects and drug-related toxicity.

Our work is limited by the study design; an observational, longitudinal study design could better inform the study objectives than a cross-sectional survey. Nevertheless, questionnaires are a robust research technique that allow data collection without direct interaction between researchers and respondents [43]. Another factor limiting the quality of the results is the fact that the survey is not formally validated (although one patient reviewed it). This implies that some questions could have been misleading and confusing for respondents. In order to avoid this, a deep quality check of answer was implemented and the results were based only on complete questionnaires and consistent answers. Recall bias could have also limited the quality of results. In order to contain it, we addressed the survey compilation suggested to the respondents, at the beginning of the data collection, the potential interruption of the survey (answers were recorded), and the need to use the same device (smartphone, iPhone, mobile) during the data collection. This should have facilitated certainty about monthly activity reporting, based on their routinely actions. In order to increase robustness, we also asked participants to answer based on their experience before the pandemic. Finally, although the average sample age (42 years) is aligned with the most recent Italian data [38], the HRQoL results are limited to the oldest TDT population. However, this limit might paradoxically represent a strength of our results; that is, we can indeed report that, if TDT Italian patients’ life expectancy can exceed the age of fifties as reported from the experts, HRQoL in persons who are treated from around forty years, like in our sample, is still deeply impaired by the disease. An additional limitation is that the study does not assess the correlation between well-being, QoL, and therapy adherence. Indeed, the relation among these factors is still unclear. It has often been shown that psychosocial factors relate to medication adherence in other pathologies [44,45,46]. In β-thalassemia, it has been observed that lower HRQoL in the physical domain is among the predictors of lower adherence to iron chelation therapy (Trachtenberg, 2014). More recently, Vosper et al. suggested that situational psychological factors are relevant for chelation adherence. Further investigations are required to explain this correlation, mostly if considering that anxiety, well-being, and vitality affect the extent of consider demanding or not managing drug therapies from patients, as our analysis suggests (please see Appendix B, Table A2, Table A3 and Table A4). Studying this aspect could be relevant as drug dosing and scheduling correlate to adherence [44,45,46,47,48,49].

## 5. Conclusions

In conclusion, this study provides new information on HRQoL in Italian TDT patients within patient advocacy organizations. Their HRQoL is greatly impacted by the disease, in terms of both physical health and psychological well-being and moderate distress as compared with the general population. Specifically, the results from different scales (SF-36 and PGWBI) both agree that vitality and general health are significantly impaired. Burden of disease is perceived as high from patients in terms of time spent to follow-up transfusions and related therapies, and thus impacted personal daily life. This finding was most impactful in terms of perceived burden of iron chelation therapy and its long-term sequela of complications, which cause fear and worries for the future. In addition, the patient survey highlights some unsolved issues in TDT patients’ management, like the jeopardized setting of care (specialized versus not specialized), center issues during transfusion sessions (like delays in the blood bags availability), and patients feeling unsupported in their care. Together, these outcomes worsen the burden of disease in TDT Italian patients and limit treatment adherence. Consequently, this work demonstrates that the current TDT management is effective in immediately saving patients’ lives by means of transfusions and iron chelation therapy, but margins of improvement in the pattern of care are evident, if considering a holistic perspective of disease management and unsolved management issues. These findings taken together exhibit the necessity and opportunity for innovative therapeutic approaches, like erythroid maturation agent and gene therapy.

## Figures and Tables

**Figure 1 jcm-11-00015-f001:**
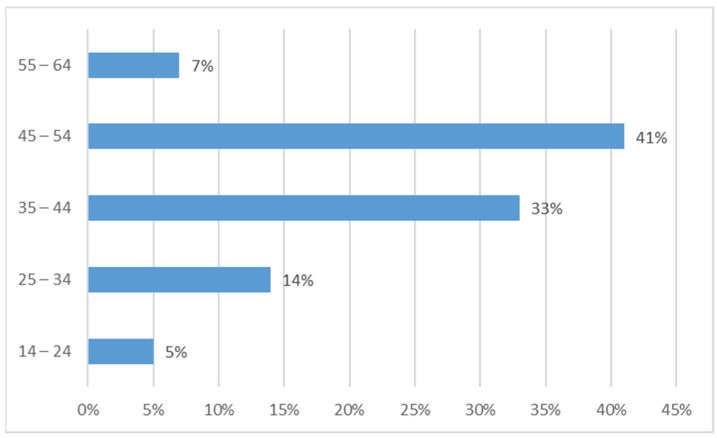
Distribution of sample age.

**Figure 2 jcm-11-00015-f002:**
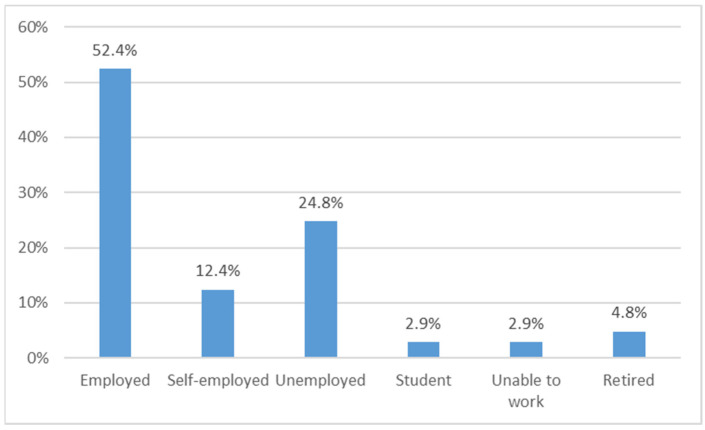
Distribution of employment status.

**Figure 3 jcm-11-00015-f003:**
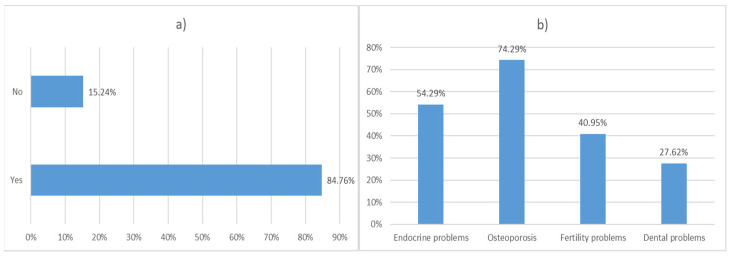
*β*-thalassemia Transfusion-dependent comorbidities. (**a**) Distribution of patients with at least one comorbidity. (**b**) Frequency of comorbidities.

**Figure 4 jcm-11-00015-f004:**
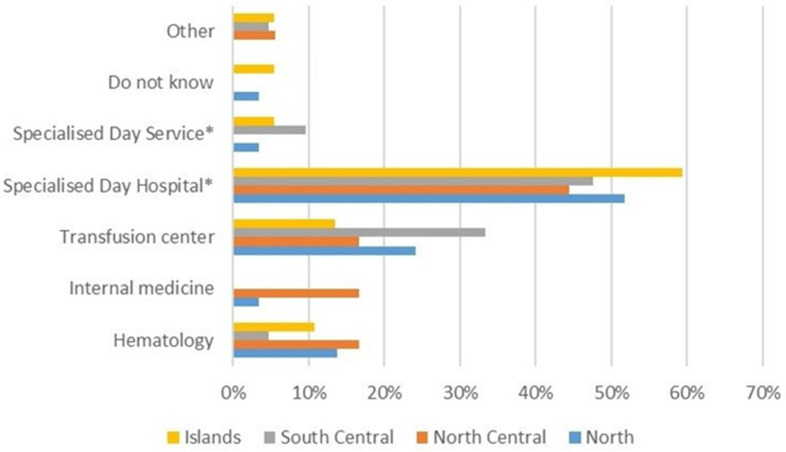
Geographic distribution of transfusions centers. The percentage is computed with respect to the total centers with similar features. * Thalassemia and hemoglobinopathies specialized centers.

**Figure 5 jcm-11-00015-f005:**
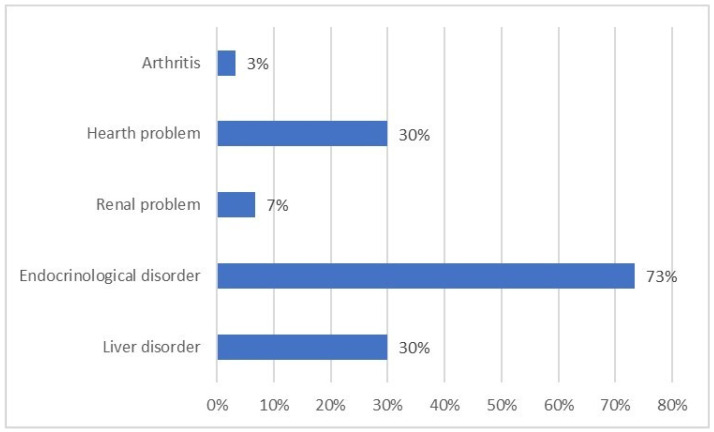
Complications of iron overload in β−thalassemia transfusion-dependent patients.

**Figure 6 jcm-11-00015-f006:**
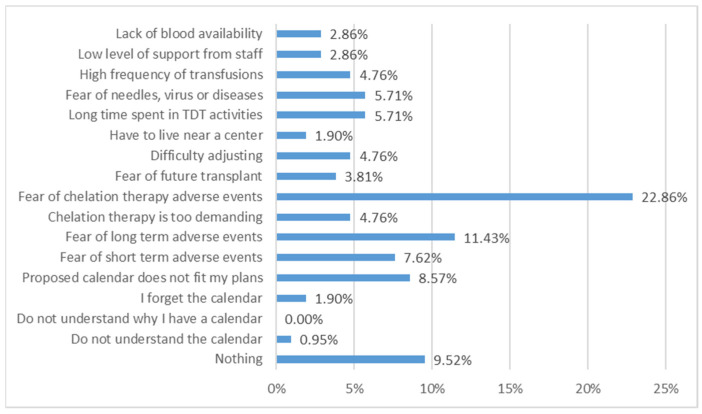
Reasons for patients’ perceived burden of their transfusion schedule.

**Figure 7 jcm-11-00015-f007:**
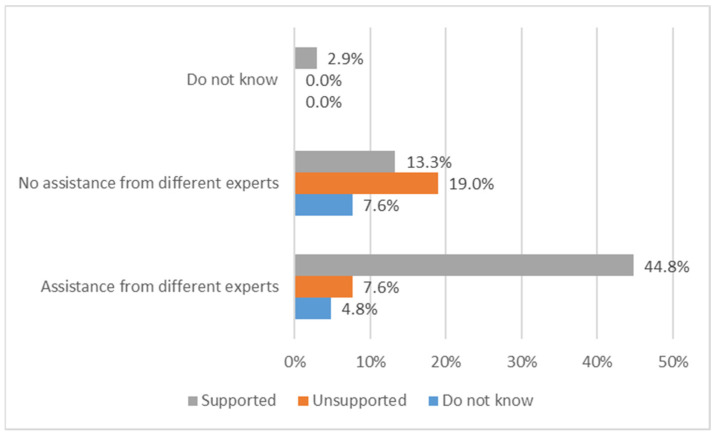
Distribution of patients reporting perception of support by its own center, depending on the presence of a multidisciplinary team.

**Figure 8 jcm-11-00015-f008:**
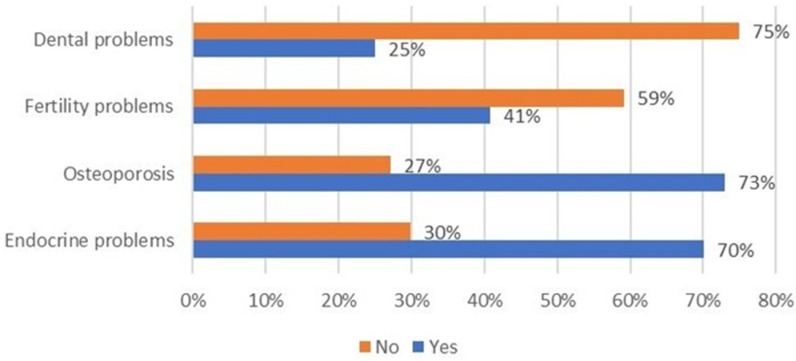
Distribution of patients reporting perception of support by their center in the comorbidities’ management.

**Figure 9 jcm-11-00015-f009:**
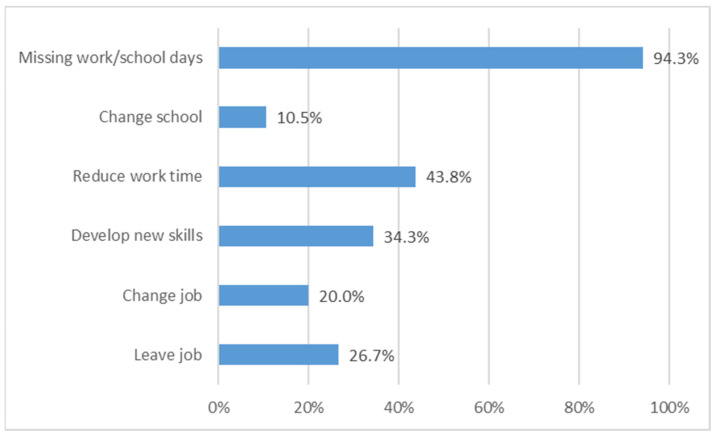
Impact of beta thalassemia on daily occupational activities.

**Figure 10 jcm-11-00015-f010:**
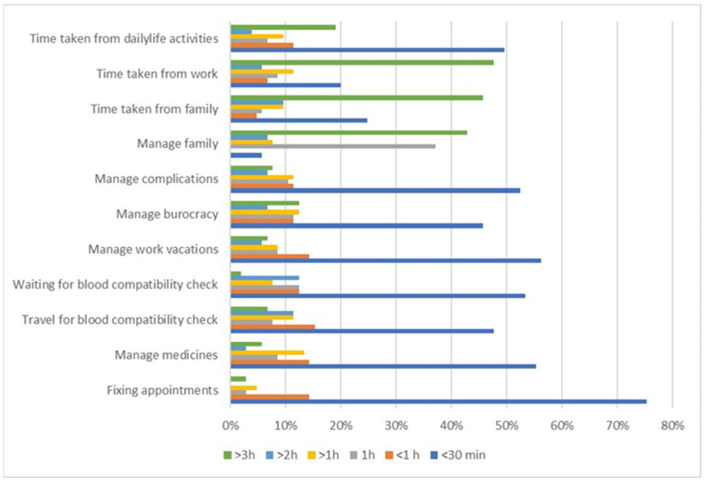
Time spent (hour/month) for transfusion-related activities on non-transfusion days: activity details.

**Figure 11 jcm-11-00015-f011:**
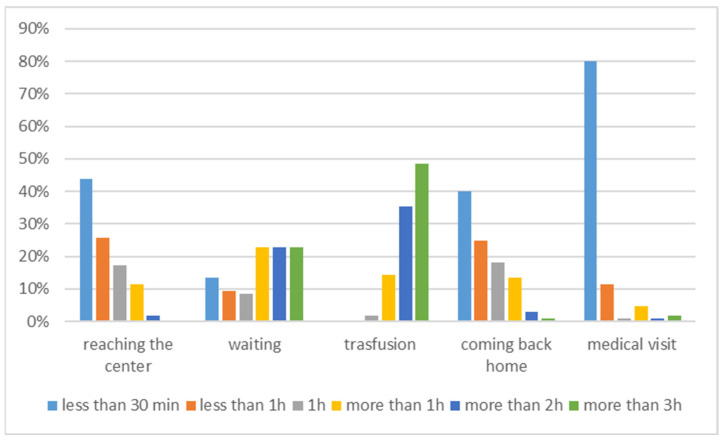
Time spent (hour/month) for transfusion-related activities during transfusion days: activity details.

**Table 1 jcm-11-00015-t001:** SF-36 T-test results. Negative point difference means reduced quality of life with respect to the general population.

Domain	Mean (SD)	Points of Difference	*p*-Value	95% Confidence Interval	
Physical functioning	74.47 (20.67)	−9.99	<0.0001	−14.08	−5.89
Role limit. due to physical health	63.33 (40.88)	−14.88	0.0004	−22.94	−6.81
Role limit. due to emotional problems	70.47 (38.20)	−5.69	0.1389	−13.25	1.87
Energy/fatigue	53.19 (21.11)	−8.7	0.0001	−12.88	−4.52
Emotional well-being	67.35 (19.77)	0.76	0.7024	−3.16	4.69
Social functioning	67.38 (26.43)	−10.05	0.0002	−15.26	−4.83
Bodily pain	65.14 (25.85)	−8.53	0.0013	−13.67	−3.38
General health	41.66 (17.71)	−23.56	<0.0001	−27.11	−20.00

**Table 2 jcm-11-00015-t002:** PGWBI T-test results. Negative point difference means reduced quality of life with respect to the general population.

Domain	Mean (SD)	Points of Difference	*p*-Value	95% Confidence Interval	
Anxiety	16.34 (4.83)	−0.96	0.0542	−1.93	0.017
Depression	12.35 (2.86)	−0.05	0.85	−0.56	0.467
Well-being	11.16 (3.89)	−0.64	0.1104	−1.42	0.148
Self-control	11.41 (2.94)	−0.39	0.1929	0.98	0.199
General health	9.49 (2.89)	−1.61	<0.0001	−2.19	−1.021
Vitality	11.14 (3.92)	−2	<0.0001	−2.79	−1.207

**Table 3 jcm-11-00015-t003:** Time spent (in hour/month) for transfusion-related activities in non-transfusion days.

	Therapy Management ^1^	Family Management ^2^	Total Time
Median	2	1	4
1° quartile	5	2	10
3° quartile	11	8	20
Mean	11.7	4.92	16.62
Stand. Deviation	20.52	5.31	22.9
Range	0–150	0–20	1–170

^1^ Therapy management = manage agenda, manage complications, manage bureaucracy, waiting for checks. ^2^ Family management = coordinating baby-sitting for children and old persons in the family, organizing vacations.

## Data Availability

The datasets generated during and/or analyzed during the current study are available from the corresponding author on reasonable request.

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
