# Peer review of "Quality of Life and Burden of Disease in Italian Patients with Transfusion-Dependent Beta-Thalassemia"

_jcm, 2021, doi:10.3390/jcm11010015_

Round 1
Reviewer 1 Report
Authors performed a good study analyzing quality of life of patients with transfusion-dependent beta-thalassemia.
Major comments:
- Title should be "...in Italian patients with transfusion-dependent beta-thalassemia".
- Abstract. The conclusion of the abstract can not be achieved after reading the body of the abstract.
- Discussion. This reviewer believes that this section is too large. I would recommend authors to focus in discussing the results observed in the stuy and not repeating the results. For example, second paragrap of the discussion can be removed entirely.
Minor comments:
- Abstract. Authors should report age as median (interqurtile range (IQR)).
- Abstract. The sentence beginnig "Well-being as assessed by total PGWBI score,..." is confusing.
- Introduction. The starting sentence shoud be "Transfusion-dependent beta-thalassemia (TDT)..."
- Material and methods. Were included only patients aged 14 years? Or >=14 years?
Author Response
We thank the reviewer for the useful comments. Following our point-by-point response to the reviewer's comments:
MAJOR COMMENTS
1) We agree with the reviewer and have modified the title
2) We have modified the abstract to better clarify the results and conclusion of the study.
3) We agree with the reviewer for the second paragraph (we have deleted it in the revised version of the manuscript). We believe the rest of the discussion is necessary to emphasize the value of the results with respect to the literature.
MINOR COMMENTS
1) We agree with the reviewer and modified the abstract accordingly.
2) We agree with the reviewer and modified the abstract accordingly.
3) We agree with the reviewer and modified the introduction accordingly.
4)The study included all patients >=14. We have clarified it in the revised version of the manuscript.
Reviewer 2 Report
This is an interesting, carefully performed, and generally well written manuscript. It provides an update on quality of life in disease burden in Italian beta thalassemia patients, updating and expanding upon the previous report on this topic some dozen years ago.
Comments relate mostly to clarification.
- It would be helpful if the authors would comment in a little more detail about the potential benefit of chelation, namely the adverse effects of transfusional iron overload and the potential adverse effects of increased iron absorption as a results of the suppression of Hepcidin by ineffective erythropoiesis.
- The authors make the interesting observation that chelation in itself contributes to the burden of disease in roughly 25% of cases. Approximately 20% of patients are on subcutaneous/parental chelation therapy, which is typically perceived as problematic by patients. There should be some discussion as to whether the patients perceiving chelation burden are the same as those receiving parenteral/subcutaneous chelation, and any differences between the subcutaneous chelation group and the oral chelation group in quality-of-life/disease burden should be commented upon. If the study mechanism does not allow those patients to be separated, then that should be noted specifically as a limitation of the study.
- The authors need to be more clear about what they mean by the term “naturalistic”. Do they mean “replicating actual patient circumstances” or (to use a phrase sometimes employed in this circumstance) “real world”? Do they perceive this study as falling into the category sometimes called a “pragmatic” clinical trial?
Author Response
We thank the reviewer for the positive comment and the suggestions to make the manuscript clearer. Following, our point-by-point response to the reviewer's comments:
1) We agree with the reviewer and have modified the introduction accordingly
2) The chelation burden is statistically lower for patients following oral therapy when compared to patients following subcutaneous therapy. No effect on the quality of life/wellbeing was noted. We modified the manuscript (section 3.4) accordingly
3) By "naturalistic" we mean that the study could directly give voice to TDT patients, without any intermediated selection from clinical centers to analyze the real heterogeneity of the setting of care. We modified the manuscript accordingly.